# Zebrafish fast muscle contractions avoid the mammalian requirement for voltage-gated Na⁺ channels

Léa Demesmay[1]☯, Romane Idoux[1]☯, Christine Berthier[1], Claire Bernat[2,3], Léon Espinosa[4], Vincent Jacquemond[1], Frédéric Brunet[5], Angel Maunier-Mercier[2], Philippe Lory[2,3], Sophie Nicole[2,3‡]*, Bruno Allard[1‡]*

**1** Physiopathologie et Génétique du Neurone et du Muscle, CNRS 5261, INSERM U1315, Université Claude Bernard Lyon 1, Lyon, France, **2** IGF, Univ Montpellier, CNRS, Inserm, Montpellier, France, **3** LabEx 'Ion Channel Science and Therapeutics', Montpellier, France, **4** Laboratoire de chimie bactérienne, CNRS 7283, Aix-Marseille Université, Marseille, France, **5** Institut de Génomique Fonctionnelle de Lyon, ENS Lyon, CNRS 5242, INRA USC1370, Université Claude Bernard Lyon 1, Lyon, France

☯ These authors contributed equally to this work.
‡ SN and BA also contributed equally to this work.
* sophie.nicole@igf.cnrs.fr (SN); bruno.allard@univ-lyon1.fr (BA)

## Abstract

Fast skeletal muscle fibers from zebrafish share a number of functional properties with mammalian twitch muscle fibers, making this vertebrate a precious model to investigate the pathophysiology of neuromuscular disorders. We previously reported that action potentials (APs) from zebrafish fast fibers exhibit low amplitude and require unusually strong negative resting membrane voltage levels to be elicited. In this study, using voltage-clamp and current-clamp techniques, we explored the properties of voltage-gated Na⁺ channels (Na$_V$) responsible for initiation and propagation of AP in isolated adult zebrafish fast skeletal muscle fibers and compared them to mouse fast-twitch muscle fibers using the same experimental conditions. We found that kinetics of activation and inactivation of Na$_V$ were faster in zebrafish fibers and, overall, that the voltage-dependence of inactivation was shifted by 25 mV toward negative voltages as compared to mouse fibers, yielding a mean half-inactivation potential of −90 mV. In agreement with these findings, recording of APs at various resting membrane potentials indicated that APs vanished for resting membrane potentials less negative than −80 mV in zebrafish, whereas APs could still be elicited from resting membrane potentials as low as −60 mV in mice. In addition, Ca²⁺ transients induced by field stimulation were insensitive to Na⁺ current blockade in zebrafish but not in mouse fibers. Fluorescence labeling of nicotinic acetylcholine receptors showed that zebrafish fast fibers were multi-innervated with a mean distance between extra-synaptic sarcolemma and motor endplates of 14 μm, expected to lead to negligible attenuation of depolarization propagated from endplates. Finally, knock out of the two genes encoding pore-forming Na$_V$ subunits in zebrafish muscles

**Data availability statement:** All relevant data are within the paper and its Supporting information files or available at https://doi.org/10.5281/zenodo.17290038.

**Funding:** This work was supported by the Centre National de la Recherche Scientifique (UMR 5261 to LD, RI, CB1, VJ, BA; UMR 5203 to CB2, AM, PL, SN; UMR 7283 to LE; UMR 5242 to FB), the Institut National de la Santé et de la Recherche Médicale (U1315 to LD, RI, CB1, VJ, BA; U1191 to CB2, AM, PL, SN), the Institut National de Recherche pour l'Agriculture, l'alimentation et l'Environnement (USC 1370 to FB), the Université Claude Bernard Lyon 1 (PGNM to LD, RI, CB1, VJ, FB, BA), the Université de Montpellier (IGF to CB2, AM, PL, SN), the Université Aix-Marseille (LCB to LE), the Ecole Normale Supérieure de Lyon (IGFL to FB), the Association Française contre les Myopathies (AFM)-Téléthon (MyoNeurALP2 to LD, RI, CB1, VJ, BA ; grant 23677 to CB2, AM, PL, SN), the Fondation Maladies Rares (FONDATION_SAM-RD – I20141211 CB2, AM, PL, SN) and the Agence Nationale pour la Recherche (Programme Laboratoires d'Excellence, ICST ProjetIA-11-LABX-0015 to CB2, PL, SN). LD was supported by a post-doctoral fellowship from AFM -Téléthon. RI was supported by a PhD student fellowship from the Fondation pour la Recherche Médicale (FRM). AM received an engineer's salary from AFM -Téléthon. The funders had no role in study design, data collection and analysis, decision to publish, or preparation of the manuscript.

**Competing interests:** The authors have declared that no competing interests exist.

**Abbreviations :** AChRs, acetylcholine receptors; APs, action potentials; DHPRs, dihydropyridine receptors; Enu, ethyl-N-nitrosourea; EZRC, european zebrafish research center; $MeSO_3$, methanesulfonic acid; $Na_V$, voltage-gated $Na^+$ channels; PBS, phosphate-buffered saline; RyRs, ryanodine receptors; SR, sarcoplasmic reticulum; TEA, tetraethylammonium-OH; TTX, tetrodotoxin; WGD, whole-genome duplication; WT, wildtype; ZFIN, zebrafish information network.

did not induce any change in locomotion and escape behavior of the animals. Taken together, these data question the role of $Na_V$ and the occurrence of APs in zebrafish fast muscle.

## Introduction

Zebrafish is a valuable vertebrate model system for investigating the pathophysiology of neuromuscular disorders [1]. Indeed, over 70% of genes associated with disease states in humans have a functional ortholog in zebrafish [2]. Fast skeletal muscle fibers from zebrafish trunk, which are recruited for high-speed swimming, are by far the most used model to explore muscle pathophysiology because these fibers share a large number of common functional properties with mammalian twitch skeletal muscle fibers. Indeed, zebrafish fast skeletal muscle fibers are innervated by cholinergic motor neurons, and are described as able to fire action potentials (APs) spreading along the transverse tubular system to activate dihydropyridine receptors (DHPRs) that induce opening of the sarcoplasmic reticulum (SR) $Ca^{2+}$ channels, the ryanodine receptors (RyRs), in charge of the release of $Ca^{2+}$ ions activating contraction [3–6]. However, all the steps from excitation to SR $Ca^{2+}$ release display some differences in zebrafish muscle as compared to mammals. In contrast to mammalian skeletal twitch muscle fibers, fast skeletal muscle fibers from adult zebrafish are multi-innervated with several motor endplates per fiber [7] and DHPRs solely act as voltage sensors controlling the opening of RyRs since they do not possess the $Ca^{2+}$-conducting function of mammalian voltage-dependent L-type $Ca^{2+}$ channels [8]. Additionally, the kinetics of AP repolarization, the kinetics of intramembrane charge movements resulting from the depolarization-induced changes in the configuration of DHPRs and the kinetics of the decay phase of SR $Ca^{2+}$ release are all significantly faster in zebrafish muscle fibers as compared to mice [6]. Concerning APs, other relevant differences with mammals have been reported in muscle fibers from adult zebrafish. First, the average amplitude of APs was found to be significantly reduced and barely overshooting in isolated fibers from zebrafish fast skeletal muscles as compared to mice under the same experimental conditions [6]. More intriguingly, it was observed that APs could be hardly elicited when the resting membrane potential was set to values less negative than −100 mV [6]. Yet, the average resting membrane potential of fast muscle fibers measured with an intracellular microelectrode in situ in zebrafish was found to be −82 mV [7], thus questioning whether fast skeletal muscle fibers are able or not to fire APs under physiological conditions.

Taken together, these data prompted us to explore the biophysical properties of voltage-gated $Na^+$ channels ($Na_V$) which are responsible for initiation and propagation of AP using voltage- and current-clamp techniques in isolated fast skeletal muscle fibers from adult zebrafish and to compare them to properties of mouse $Na_V$ using the same experimental conditions. The main finding of our study is that the voltage inducing half-inactivation and the reversal potential of $Na_V$ currents are 25 and 11 mV more negative, respectively, in zebrafish fibers as compared to mouse fibers. In agreement with these findings, we showed that overshooting APs could not be fired in zebrafish

muscle from resting membrane potentials less negative than −80 mV, whereas in mouse muscle, overshooting APs could still be elicited from resting membrane potentials as low as −60 mV. In addition, $Ca^{2+}$ transients induced by field stimulation were insensitive to $Na^+$ current blockade by tetrodotoxin (TTX) in zebrafish but not in mouse fibers. Labeling of nicotinic acetylcholine receptors (AChRs) in isolated fibers from zebrafish fast skeletal muscle indicated that the maximal distance between adjacent motor endplates is so short that this brings into question the necessity for zebrafish fibers to fire and propagate APs. In agreement with this assumption, knock out of the two genes encoding pore-forming subunits of $Na_V$ expressed in zebrafish fast skeletal muscles did not induce any detectable motor phenotype.

## Results

A first series of voltage clamp experiments was performed on isolated fibers from zebrafish fast muscle to record voltage-activated currents underlying the rising and decay phase of APs in the presence of external Tyrode solution. Fig 1A shows that depolarizing pulses of increasing amplitudes from a holding potential of −110 mV elicited an early inward current that inactivated with time, followed by a delayed outward current, that were unambiguously identified as voltage-gated $Na^+$ and a mixture of $K^+$ and $Cl^-$ currents, respectively. Fig 1B indeed shows that exposition of the fiber to an external solution containing 1 μM TTX and an external solution containing 20 mM TEA and the impermeant anion $MeSO_3^-$ substituted to $Cl^-$ totally suppressed the early inward and the delayed outward current, respectively.

Activation and inactivation properties of $Na^+$ currents were then investigated in zebrafish muscle fibers using a two-pulse protocol in the presence of the external solution blocking delayed outward currents and from a holding membrane potential of −110 mV (Fig 2A). In the vast majority of fibers, maximal $Na^+$ current elicited by the first pulse could not be recorded since $Na^+$ currents reached amplitude larger than the 100 nA that the voltage clamp amplifier could maximally inject. Consequently, the current-voltage relationship could not be determined over the entire range of voltages. However, $Na^+$ currents could still be recorded at voltage threshold value and in response to depolarizing pulses inducing

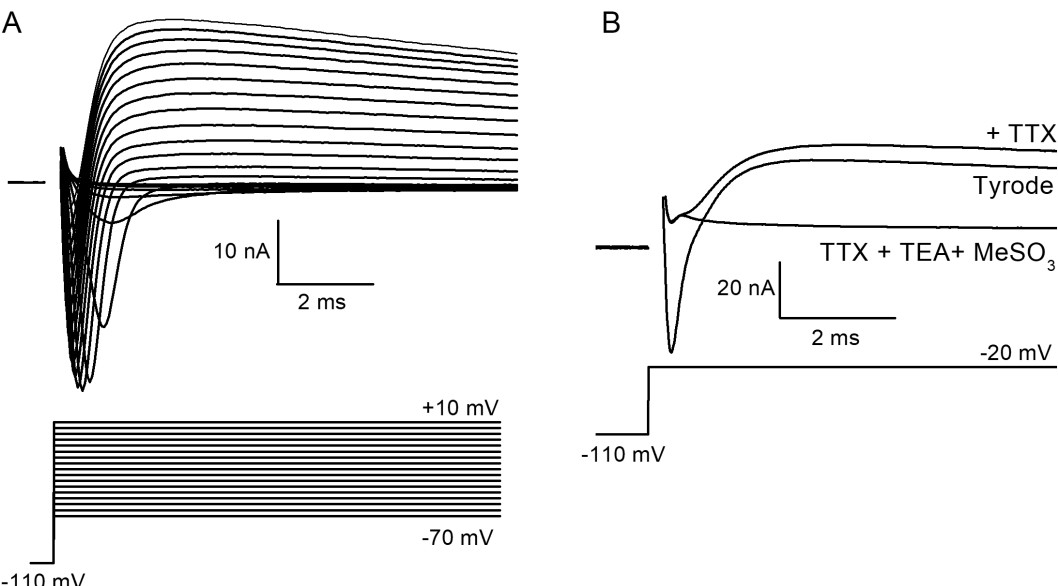

**Fig 1. Voltage-activated currents in isolated adult zebrafish muscle fibers. (A)** Voltage-gated early $Na^+$ and delayed $K^+/Cl^-$ currents (upper traces) evoked by depolarizing pulses of increasing amplitudes (lower traces) in the presence of an external Tyrode solution. **(B)** Effect of external solutions containing 1 μM tetrodotoxin (TTX) or 1 μM TTX with 20 mM TEA and in which external $Cl^-$ has been replaced by $MeSO_3^-$ on currents elicited by a depolarizing pulse to −20 mV in the same fiber.

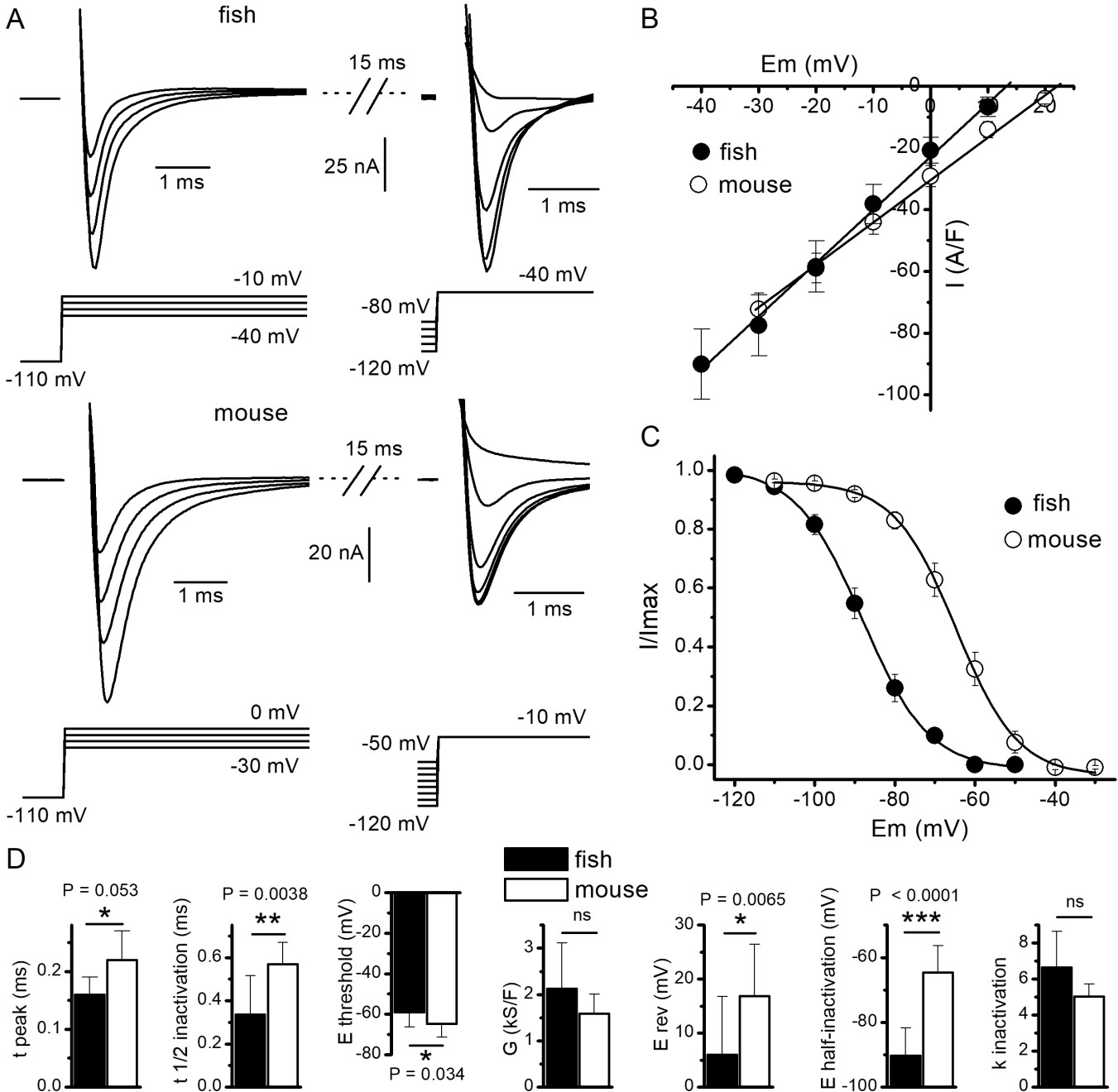

**Fig 2. Activation and inactivation properties of voltage-activated Na⁺ currents in zebrafish and mouse muscle fibers. (A)** Voltage-activated Na⁺ currents elicited in a zebrafish (upper panel) and in a mouse (lower panel) muscle fiber in response to a two-pulse protocol consisting in a first 20 ms voltage pulse of increasing amplitude followed by a second 20 ms voltage pulse given to −40 mV in zebrafish and to −10 mV in mouse in the presence of the external TEA+MeSO$_3$ containing solution. **(B)** Relationship between mean amplitude of peak Na⁺ current and membrane potential in 21 fibers from 6 zebrafish and 26 fibers from 5 mice. Because of limitation of current injected by the amplifier for voltage pulses from threshold to −40 mV in zebrafish or to −30 mV in mice, only the linear part of the relationship is presented from these voltages to reversal potentials. In the graph, mean datapoints were fitted using a linear equation with values for conductance and $E_{rev}$ of 1.7 kS/F and +13 mV, and 1.36 kS and +22 mV, in zebrafish and in mouse, respectively. **(C)** Relationship between the mean relative amplitude of the Na⁺ current elicited by the second voltage pulse and the applied voltage in the first pulse in zebrafish and in mouse. In the graph, mean datapoints were fitted using a Boltzmann equation with values for $V_{0.5}$, and $k$ of −88 mV and 8.3 mV, and −64 mV and 7.7 mV, in zebrafish and in mouse, respectively. **(D)** Parameters of activation and inactivation of voltage-activated Na⁺ currents in zebrafish and in mouse muscle fibers. Time to peak ($t_{peak}$) corresponds to time elapsed between full charge of membrane capacitance and peak of Na⁺

current. Time to half inactivation ($t_{1/2\ inactivation}$) corresponds to time elapsed between Na$^+$ current peak and 50% inactivation of the current. Other parameters have their usual definition. The indicated *P* values were calculated using a statistical nested *t* test (see Materials and methods). The error bars correspond to SD values that were calculated from pooled values obtained in 21 fibers from 6 zebrafish and in 26 fibers from 5 mice. The data underlying Fig 2D can be found in S1 Data.

Na$^+$ currents with amplitude lower than 100 nA which further decreased linearly with increasing depolarizations. In these conditions, we found that the average threshold for activation of voltage-gated Na$^+$ currents was −59 ± 7 mV. Plotting in each cell the peak Na$^+$ current amplitude as a function of voltage over a range for which the current decreased linearly with voltage and fitting datapoints with a linear relationship in each cell yielded a mean maximal Na$^+$ conductance and a mean reversal potential of 2.13 ± 0.98 kS/F and 6 ± 11 mV, respectively (Fig 2B and 2D). Recording of Na$^+$ currents in response to the second pulse given to −40 mV allowed to determine their voltage-dependence of inactivation. Fitting a Boltzmann equation to the relationship between the relative amplitude of the second pulse-induced Na$^+$ current and voltage reached in response to the first pulse yielded a mean voltage inducing half-inactivation and a steepness factor of −90 ± 8 mV and 6.7 ± 2 mV, respectively (Fig 2A, 2C, and 2D). Additionally, we found that the average time for Na$^+$ current to peak was 160 ± 30 μs in response to a voltage pulse given to −20 mV and the average time from Na$^+$ current peak to half-inactivation was 336 ± 180 μs (Fig 2D).

The same experiments were carried out on isolated fibers from mouse skeletal muscle using the same experimental conditions and protocols used on zebrafish fibers (Fig 2A). This time, in all the mouse muscle fibers investigated, maximal Na$^+$ currents elicited by the first pulse could never be recorded since their amplitude always exceeded 100 nA. Over the voltage range the Na$^+$ current could be recorded, fitting a linear relationship between peak Na$^+$ current amplitude and voltage yielded a mean maximal Na$^+$ conductance of 1.58 ± 0.42 kS/F, not significantly different from the one found in zebrafish (Fig 2B and 2D). The voltage threshold for Na$^+$ current activation was −65 ± 6 mV, significantly more negative than the one found in zebrafish (Fig 2D). In contrast, the average reversal potential (17 ± 10 mV) was significantly more positive than the one measured in zebrafish (Fig 2B and 2D). The most marked differences were the mean voltage inducing half-inactivation equal to −65 ± 8 mV (Fig 2C), which was very significantly more positive than the one found in zebrafish (−90 ± 8 mV), and the time to peak (220 ± 50 μs) and the time from Na$^+$ current peak to half-inactivation (570 ± 10 μs), which were both significantly longer than the ones found in zebrafish (Fig 2D).

The important shift toward negative voltages of the voltage-dependence of Na$^+$ current inactivation in zebrafish may explain why the fraction of available Na$^+$ channels is too small for APs to be elicited from resting membrane potentials less negative than −95 mV, as observed previously [6]. In order to determine the least negative resting membrane potential that can be set to allow firing of APs in zebrafish fibers, APs were evoked in current clamp conditions from a resting membrane potential of −110 mV, which was then further made less negative until APs could not be elicited anymore (Fig 3A). Fig 3B shows that, in average in zebrafish, the spike amplitude diminished when the resting membrane potential was set to values less negative than −95 mV, APs either no more overshot or vanished for resting membrane potentials less negative than −85 mV and systematically vanished for resting membrane potential less negative than −70 mV. In contrast, in mouse muscle fibers, the spike amplitude did not change until the resting membrane potential became less negative than −75 mV, still overshot for resting membrane potentials set at −60 mV and 80% of fibers could still fire APs for resting membrane potentials set to voltages as low as −50 mV (Fig 3A and 3B). Comparison of the maximal spike amplitude in zebrafish (+7 mV) and in mouse (+25 mV) elicited from resting membrane potential set at −100 mV and −80 mV, respectively, confirmed the significant lower value previously found in zebrafish (nested *t* test, *P* = 0.018) [6]. Finally, we performed a series of in situ intracellular microelectrode recordings to measure the resting membrane potential of fast muscle fibers located in the deep region of trunk muscle using an external solution containing 2 mM K$^+$ (S1 Fig). The mean resting membrane potential of fast zebrafish fibers was found to be −78 ± 0.8 mV (124 fibers from 4 fish). Taken together, our data

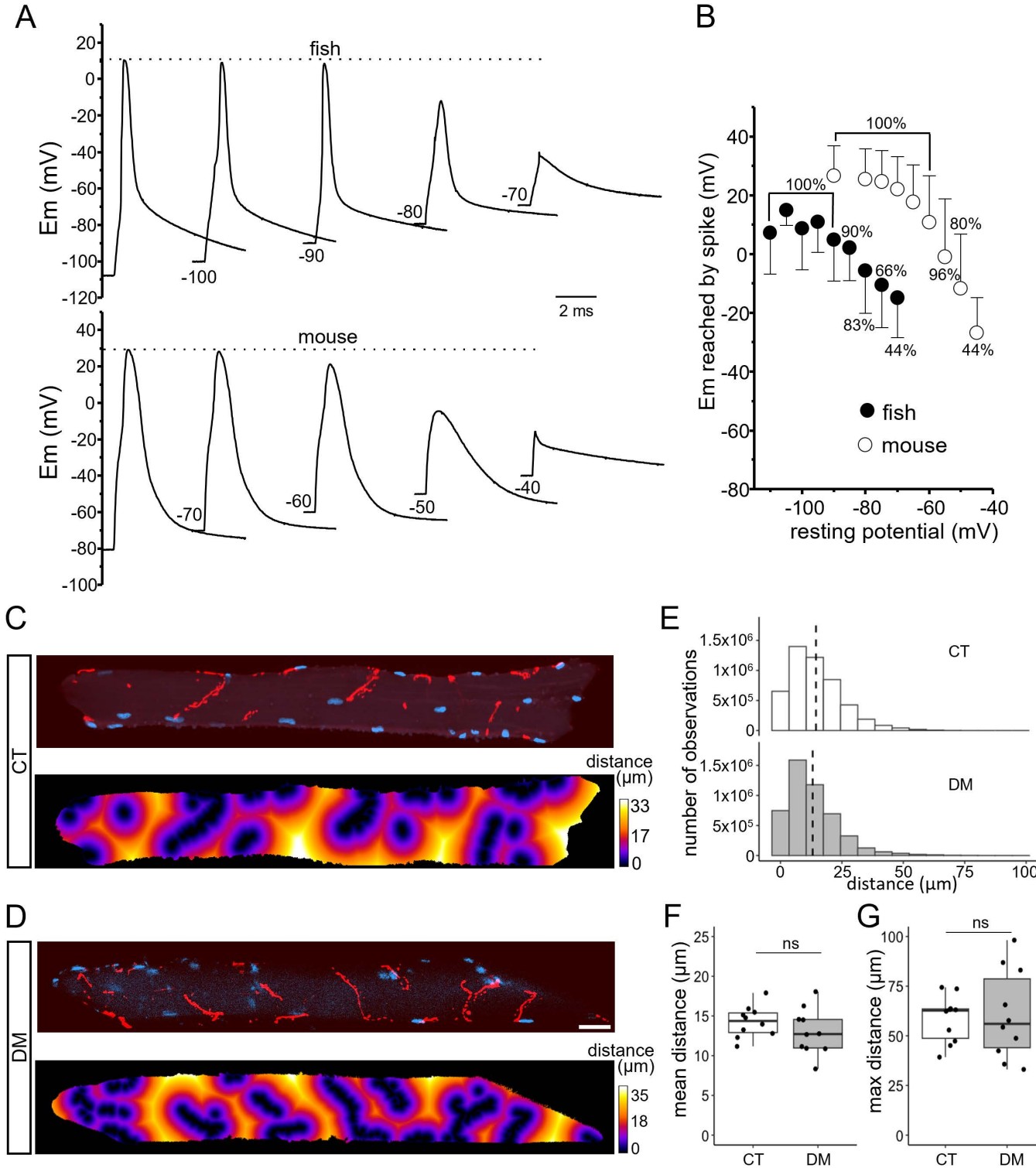

**Fig 3. Effect of lowering resting membrane potential on action potentials (APs) in zebrafish and mouse muscle fibers, and distance analysis of acetylcholine receptors (AChRs) clusters on isolated muscle fibers from control and *scn4aa^{-/-};scn4ab^{-/-}* mutant zebrafish. (A)** APs were elicited by applying 0.5 ms supraliminar depolarizing currents from various resting membrane potentials set by injecting constant currents in a current-clamped zebrafish (upper panel) and mouse muscle fiber (lower panel). **(B)** Relationship between spike amplitude and resting membrane potentials in

18 fibers from 9 fish and in 25 fibers from 4 mice. Each datapoint corresponds to the mean values of voltage reached by the spike. The percentage of fibers firing APs is indicated for each resting potential. The error bars correspond to SD values that were calculated from pooled values obtained in fish and mice, respectively. The data underlying Fig 3B can be found in S1 Data. **(C, D)** AChRs and nuclei labeling with α-bungarotoxin (red) and Hoechst (blue), respectively, in fast muscle fibers from control (CT) and mutant fish (DM). Measurements were made on Z-stacks images of whole fibers by dividing each fiber at its equator and applying Z-projection to each half. Z-projections of the upper half of a representative CT and DM fiber are shown (upper panels). Euclidean distance transformation of half-fiber Z-projection generated a distance mapping visualized by colored gradient (lower panels). Bar = 30 μm. **(E)** Euclidean distances corresponding, for each point of the extra-synaptic sarcolemma, to the shortest distances to the nearest AChRs clusters were collected, and histogram of the distribution of pooled data of 10 fibers from 3 control (top) and 10 fibers from 4 mutant fish (bottom) is shown. The dashed lines correspond to mean values. **(F)** Box plots of mean distance in control (CT) and in mutant (DM) fibers. **(G)** Box plots of maximal distance in control (CT) and mutant (DM) fibers. Wilcoxon test shows the absence of significant difference for mean and maximal distances between mutant and control ($P = 0.22$ and $P = 1.0$, respectively). Data points of each fiber are shown by dots. The data underlying Fig 3E–3G can be found in https://doi.org/10.5281/zenodo.17290038.

question whether APs are elicited in zebrafish fast muscle fibers in physiological conditions since, according to our data obtained on inactivation, only 20% of $Na_V$ are available for activation at a resting membrane potential of −78 mV.

In order to further address this question, we tested whether isolated fast skeletal muscle fibers of zebrafish are able to twitch by firing APs in response to field stimulation. For this purpose, we measured changes in intracellular $Ca^{2+}$ in isolated fibers loaded with the $Ca^{2+}$ indicator indo-1 in response to 0.5-ms voltage pulses applied with field electrodes. Stimulation pulses of high voltage were found to induce a transient increase in intracellular $Ca^{2+}$, the size of which augmented with stimulation, and which was not affected by addition of 2 μM TTX in the external solution (S2 and S3 Figs). These data give evidence that the $Ca^{2+}$ transients recorded in these experimental conditions result from local electrotonic responses and not from firing all-or-none AP. In parallel, we found that stimulation pulses of much lower voltage induced $Ca^{2+}$ transients in isolated mouse muscle fibers that were completely and reversibly suppressed by addition of 2 μM TTX in the external solution (S4 Fig). These data indicate that zebrafish fast muscle fibers do not fire AP to produce $Ca^{2+}$ transients and to contract in response to field stimulation in these experimental conditions whereas mouse muscle fibers do.

In mouse, twitch muscle fibers are mono-innervated and, to produce maximal contraction, fibers must be homogeneously depolarized over their full length, which is achieved by means of firing and propagation of APs from the centrally located motor endplate. In contrast, fast zebrafish muscle fibers are known to exhibit multiple cholinergic synapses organized in short rows densely distributed over the fibers [9]. In these conditions, depending on the distance apart each motor endplate, the voltage change induced by simultaneous opening of AChRs at the level of each motor endplate might homogenously depolarize the fiber without any need to generate and propagate APs. In order to determine whether APs propagation is required, AChRs were labeled with α-bungarotoxin in isolated fast muscle fibers and the shortest distances to the nearest AChRs cluster were measured for all extra-synaptic sarcolemmal locations (Fig 3C). We first confirmed that cholinergic synapses were organized in short rows wrapping around the fibers and distributing over the entire length of the fibers. More importantly, by analyzing the sarcolemmal distribution of AChRs using Euclidean distance transformation mapping, we found that the maximal distance between each point located on the extra-synaptic sarcolemma from an AChRs cluster was in average 58.52 ± 0.12 μm, and the mean distance was 14.4 ± 0.01 μm (Fig 3E–3G).

To further determine whether activation of $Na_V$ is required for muscle contraction in vivo, we investigated the locomotor consequences of deleting $Na_V$ channels in skeletal muscles of zebrafish. Two zebrafish *scna* genes encoding a pore-forming α-subunit of $Na_V$ are expressed in skeletal muscles: *scn4aa* and *scn4ab*, which encode a protein of 1829 ($Na_V1.4aa$) and 1,784 ($Na_V1.4ab$) amino acids, respectively [10,11]. We confirmed by PCR on reverse-transcribed mRNAs that these two paralogs are expressed in skeletal muscles of adult zebrafish (S5 Fig). $Na_V1.4aa$ and $Na_V1.4ab$ share 72% identity, and 72% and 71% identity with the mouse $Na_V1.4a$ ortholog, respectively (S6 Fig and S1 Table). Divergences lie in the intracellular loops, whereas the transmembrane segments and the inactivation gate (located in the intracellular DIII– DIV loop) are extremely well conserved with the exception of S2 segments of domains I and IV (62% and 55% identity with mouse $Na_V1.4a$, respectively) (S6 and S7A Figs).

We established stable mutant lines with Ethyl-N-nitrosourea (Enu)-induced point nucleotide variations predicted to be null mutations for s*cn4aa* and *scn4ab* [12]. The s*cn4aa* variation (allele sa1270) is located in the acceptor splice site of intron 10, and replaces the nucleotide G of the invariable splice site AG sequence by an A (c.T1498-1g>a). The s*cn4ab* variation is one nonsense allele (allele sa33152, c.1393A>T) located in exon 10 and is predicted to introduce a stop codon (S7 Fig). We first established monogenic lines, and did not observe any obvious locomotor phenotype in homozygous s*cn4aa*⁻ᐟ⁻ or *scn4ab*⁻ᐟ⁻ mutant fish. We then intercrossed the monogenic mutant lines to establish a digenic *scn4aa;scn4ab* mutant line. Digenic homozygous (*scn4aa*⁻ᐟ⁻; *scn4ab*⁻ᐟ⁻) mutants were obtained at the adult age, were fertile, and did not display higher lethality compared to control fish over 18 months.

Molecular analyses of mRNAs extracted from skeletal muscles of control and mutant fish at the adult age confirmed the null effect of the *scn4aa* and *scn4ab* mutations (S7 Fig). PCR analyses done on skeletal muscle mRNAs showed that the *scn4aa* splicing mutation leads to a deletion of 10 nucleotides into *scn4aa* mRNAs due to the use of one cryptic acceptor splice site located in exon 11. The predicted consequence of the *scn4aa* mutation is then a frameshift in the first intracellular loop between domains I and II (p.Leu500Lysfs*6). The *scn4ab* mutation did not affect mRNA splicing and was present in *scn4ab* mRNAs as expected. Its consequence is a premature stop codon in the same intracellular loop than *scn4aa* mutation (p.Lys465*). The mutant Na$_V$1.4a proteins are therefore predicted to lack domains II, III, and IV, and could not be functional. Quantitative analyses did not detect reduced levels of *scn4aa* and *scn4ab* mRNAs in skeletal muscle RNA extracts from trunk skeletal muscles of adult digenic mutant fish compared to control fish (S7D Fig). Up-regulation of any *scna* gene (6 in zebrafish) expression to compensate for the lack of functional Na$_V$1.4a subunits was also not observed in mutant compared to control samples (S5 Fig). A series of electrophysiological experiments demonstrated that depolarization pulses of increasing amplitudes elicited only voltage-gated outward currents and absolutely no inward currents in all the 27 isolated skeletal muscle fibers studied from 7 digenic mutant fish (Fig 4A). These data demonstrate that Na$_V$ are not functional in fast muscles of digenic mutant fish and that no other voltage-gated channels carrying inward depolarizing currents compensate for the loss of their activity.

Digenic homozygous (*scn4a*⁻ᐟ⁻; *scn4ab*⁻ᐟ⁻) mutants did not show altered locomotion behavior compared to homozygous (*scn4aa*⁺ᐟ⁺; *scn4ab*⁺ᐟ⁺) controls obtained from the same laying (20 fish per group) during usual handling (S1 Video and S8 Fig). Fast muscles of fish are required for ultrafast swimming such as the one inducing escape response. Evaluating in more details the swim escape response depending upon fast muscle contraction confirmed the ability of the digenic homozygous adult mutants to escape an aversive stimulus: they displayed a C-bend shape followed by a very brief episode of ultrafast swim as did controls whatever the adult age studied (from 4 to 12 months) (Fig 4B and S2 Video). We therefore evaluated more precisely the escape response of mutant fish. Measuring the angle (head-trunk) to evaluate the head turn, did not show any difference between controls and mutants (minimal angle equal to 47.08° ± 11.79 and 59.46° ± 17.48, respectively, Fig 4C). Latency between first escape response movement and maximal head turn was also similar between the two groups (13 ± 5 for controls and 11 ± 5 ms for mutants). The subsequent rapid acceleration was as efficient in mutants as in controls: the calculated maximal speed (157 ± 27 cm/s for controls and 161 ± 33 cm/s for mutants) and latency to attain maximal speed (8 ± 4 ms for controls and 9 ± 3 ms for mutants) were not different. Distance covered in 40 ms from the escape response start was also not altered by the lack of functional Na$_V$ in skeletal muscles (3.8 ± 0.8 cm for controls and 3.8 ± 0.9 cm for mutants).

The absence of locomotor phenotype in mutant fish lacking functional Na$_V$ in skeletal muscle suggests that adult zebrafish does not need muscle Na$_V$ to efficiently contract. The depolarizing signal initiating excitation-contraction coupling could only result from the opening of the AChRs at the level of the multiple neuromuscular junctions. In that context, it might be envisioned that the lack of depolarizing Na⁺ inward current in mutant muscle is compensated by potentiation of the neuromuscular transmission. In order to test this possibility, we performed AChRs labeling experiments on mutant fish as performed in wildtype (WT). We found that the distribution of distances between each point located on the extrasynaptic sarcolemma and AChR clusters were very comparable in control and mutant fish. The mean and maximal

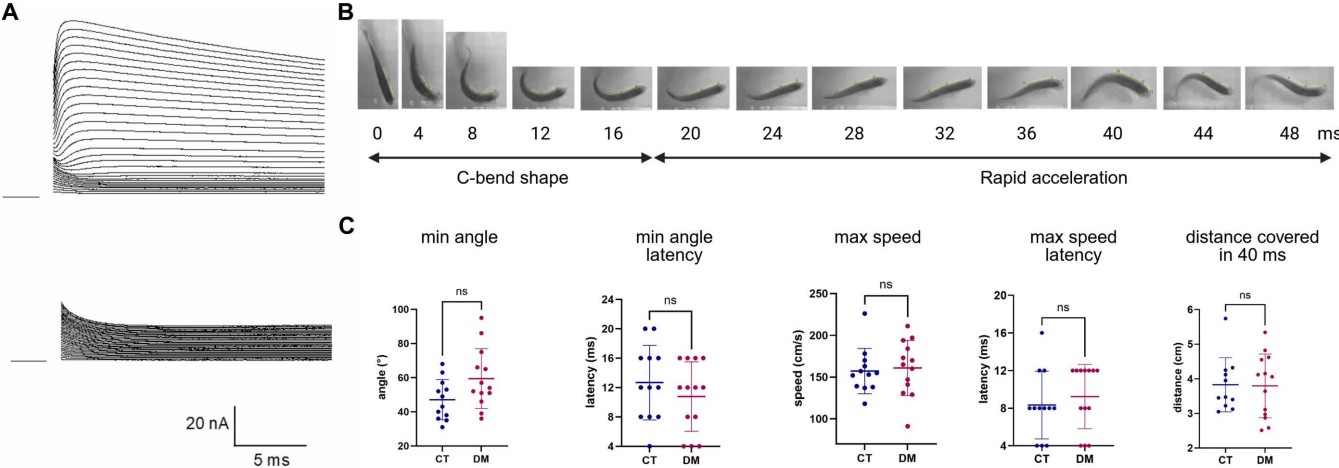

**Fig 4. Phenotype of *scn4aa*⁻ᐟ⁻; *scn4ab*⁻ᐟ⁻ fish. (A)** Absence of voltage-gated inward currents in isolated fast muscle skeletal muscle fibers of the double mutant zebrafish. Upper panel shows currents evoked by depolarizing pulses of increasing amplitudes (steps of 5 mV) from a holding potential of −110 mV applied from −100 to +40 mV in the presence of an external Tyrode solution in a mutant fiber. Lower panel shows the effect of an external solution containing 20 mM TEA and in which external Cl⁻ has been replaced by MeSO₃⁻ on currents elicited by the same voltage protocol in the same fiber. **(B)** Example of swim escape response of one mutant (*scn4aa*⁻ᐟ⁻; *scn4ab*⁻ᐟ⁻) fish. Images were extracted from the high-speed video (one frame each 4 ms). The C-bend shape occurs between the first 16 ms followed by an ultrafast swim. Yellow bars were drawn with Fiji software to calculate head-trunk angles. **(C)** Quantitative analyses of C-bend shape (minimal head-trunk angle and time to reach minimal angle) and following swim (maximal speed, maximal speed latency, and total distance covered in 40 ms following the escape response start) calculated for controls (blue) and mutants (red). No statistically significant difference was seen between the two groups whatever the parameter studied. Four- to twelve-month-old fish were studied (minimum of 11 per genotype). Graphs represent mean ± SD with individual values (unpaired Mann–Whitney tests). Created in BioRender. NICOLE, S. (2025) https://BioRender.com/6yqzet8. The data underlying Fig 4C can be found in S1 Data.

distances in control (14.4 ± 0.01 μm and 58.5 ± 0.12 μm, respectively) and in mutant fish (13.1 ± 0.01 μm and 60.6 ± 0.22 μm, respectively) were not significantly different (Fig 3C–3G). This result indicates that the absence of Na⁺ channels is not compensated by a higher density of endplates in mutant muscle.

## Discussion

To our knowledge, biophysical properties of Na$_V$ have never been explored using whole-cell voltage clamp in intact fast skeletal muscle fibers of adult zebrafish. Our study demonstrates that these properties markedly differ between zebrafish and mouse muscle under the same experimental conditions. First, kinetics of activation and inactivation of Na⁺ currents were found to be significantly faster in zebrafish muscle than in mouse muscle. This has to be related to the faster kinetics of APs, at least for the repolarization phase, that we showed in a previous study [6]. Second and main finding, the voltage dependence of inactivation is shifted by 25 mV toward negative voltages as compared to mouse, so that only 20% of Na⁺ channels would be available for firing APs from the mean resting membrane potential that we measured in situ, −78 mV, a value close to the mean value of −82 mV reported by Westerfield and colleagues [7]. This could explain why APs could be hardly elicited from resting membrane potentials less negative than −85 mV in our experiments, with a spike that never overshot. In contrast, we demonstrated that APs could still be evoked from resting membrane potentials as low as −50 mV in mouse muscle fibers, a result in close agreement with data obtained by Wang and colleagues [13] showing that APs could still be evoked in mouse muscle fibers when the resting membrane potential is brought to −60 mV by raising external K⁺. In line with this, we found that isolated mouse muscle fibers fire APs in response to field stimulation whereas zebrafish fast fibers did not and only develop electrotonic responses insensitive to TTX.

Excitability of zebrafish muscle fibers has been investigated at larval stage, but the capability of fast fibers to support APs remains controversial. Buss and Drapeau [14] were unable to evoke spikes in fast muscle fibers from zebrafish larvae, whereas spikes could be elicited by Buckingham and Ali [15], Coutts and colleagues [16], and Shimizu and colleagues [17] from a resting membrane potential as low as −70 mV. Westerfield and colleagues [7] reported that fast fibers did fire APs from an average resting membrane potential of −82 mV, but zebrafish included in the study were from 5-day to 2-year-old, and it was not specified whether APs were recorded in larvae or adult fish. In that context, one cannot unequivocally assert that zebrafish fast muscle fibers display or not the same excitable properties at larval and adult stages.

Our study also confirmed our previous findings showing that the spike amplitude was much lower in zebrafish muscle than in mouse muscle [6], even when APs were elicited from very negative resting membrane potentials, at which all $Na_V$ are available. This lower spike amplitude can be explained by the 11 mV more negative mean reversal potential of $Na^+$ currents that we found, possibly resulting from lower $Na^+$ selectivity of the channel, and/or from the persistence of a larger nonblocked $K^+$ conductance in zebrafish muscle. Finally, the voltage threshold of activation was found to be less negative in zebrafish muscle than in mouse muscle. When looking at the amino-acid sequence of the zebrafish $Na_V$1.4aa and ab regions critical for the voltage-gated channel functions, such as transmembrane segments and the inactivation gate, are highly if not strictly conserved, suggesting that the biophysical discrepancies observed are to a much greater extent due to differences in the environment of the zebrafish $Na_V$1.4 channels than in their primary structures.

Whatever the mechanisms leading to the differences we report between properties of $Na_V$ currents in zebrafish and mouse fibers, our data put into question whether APs are emitted in zebrafish fast skeletal muscle fibers under physiological conditions. This questioning is even more acute when taking into consideration multi-innervation of zebrafish fast skeletal muscle fibers. In vertebrates, innervation by multiple motor axons in adulthood is mostly restricted to tonic fibers (often named slow fibers), which are common in amphibians, reptiles, birds, and fish including zebrafish, but restricted in mammals to a subset of extraocular muscles [18,19]. These multi-innervated tonic fibers do not generate APs [20]. Mammalian phasic fibers, also named twitch fibers, usually achieve single innervation and generate and propagate APs from the motor endplate over the entire length of the fiber to induce cell-wide intracellular $Ca^{2+}$ increase and contraction [20]. In contrast, phasic fibers in teleosts are multi-innervated. As reported by others in adult zebrafish [7,9], our labeling experiments of AChRs showed multiple motor endplates distributed in short rows all along the zebrafish fast fibers. Each zebrafish adult fast fiber is innervated by a single primary motor neuron and up to three secondary motor neurons [7]. Swimming activity is mediated by the recruitment of secondary slow and intermediate motor neurons, whereas escape response results from recruitment of primary and secondary fast motor neurons [7,21]. The function of multiterminal innervation may be to allow the presynaptic motoneuron APs to reach several points of the muscle fiber faster and with greater efficiency than would be possible by conduction of APs along the sarcolemma from a single synapse [22]. Our measurements of distances between any extra-synaptic sarcolemmal location and motor endplates in zebrafish fibers agree with this assumption. Our data indeed indicate that the measured mean maximal distance is 58.5 µm, far less than the 2–3 mm length constant that has been reported for frog skeletal muscle fibers displaying tubular system organization comparable to zebrafish [23]. Assuming a 2 mm length constant ($\lambda$), the voltage change elicited at the level of a given AChRs cluster should be attenuated by a factor of $e^{-d/\lambda}$, i.e., 3% for $d$ being equal to the longest distance from AChRs clusters (58.5 µm), and by a factor of 1% for $d$ being equal to the mean distance from AChRs clusters (14.4 µm). This suggests that a regenerative and propagated response provoked by opening of $Na_V$ is not required in zebrafish muscle to homogeneously depolarize the whole fiber in a very short time. Additionally, we previously showed that membrane potential reached values close to −10 mV in response to application of ACh at the endplate of mouse muscle fibers [24] and that maximal sarcoplasmic $Ca^{2+}$ release is achieved in response to voltage pulses bringing the membrane potential around 0 mV in zebrafish muscle [6]. Considering our data showing that the mean voltage value reached by the spike from a resting potential of −80 mV did not exceed −10 mV (see Fig 3B), it can be hypothesized that the simultaneous depolarization of adjacent motor

endplates provoked by AChRs opening is large enough and as efficient as APs to quasi-maximally activate sarcoplasmic $Ca^{2+}$ release in the whole zebrafish fiber. Along this line, our data showing that the absence of $Na_V$ in skeletal muscle is not compensated by a higher density of endplates supports the assumption that multi-innervation of muscle fibers enables full activation of zebrafish fast muscle fibers without the need to open voltage-gated $Na^+$ channels.

Third strong evidence that zebrafish fast muscle fibers do not need $Na_V$ and APs to efficiently contract is that deleting the two pore-forming subunit isoforms of $Na_V$ expressed in zebrafish muscles, thereby completely suppressing voltage-gated inward current, does not produce any detectable locomotor phenotype. Although we cannot exclude that the way zebrafish were challenged during the escape test does not reproduce specific conditions of muscle contraction involving $Na_V$, our data unambiguously demonstrate that zebrafish fast muscle can contract to induce normal locomotion and escape response in the absence of $Na_V$. That may explain why the commonly used anesthetic tricaine, a $Na_V$ channel blocker, was found to not alter evoked muscle contraction in zebrafish [25]. The function of $Na_V$ channels in zebrafish is clearly different from their function in mammals, since mammalian muscle cannot contract in the absence of $Na_V1.4$ with neonatal lethality of mice and human with null mutations for this channel [26,27]. In agreement with our findings, early experiments in the fast multi-innervated white muscle of the teleost snake fish have shown that it was necessary to hyperpolarize the muscle membrane in order to obtain a spike potential with no overshoot, by direct stimulation of the muscle [28]. Additionally, repetitive stimulation of this fast muscle through its nerve evoked endplate potentials whose summation led to a single nonovershooting AP, followed by summated endplate potentials that maintained contraction during the entire period of stimulation, leading to the conclusion that the AP does not play an important role in initiating contraction in this teleost species [28].

Yet, phylogenetic and synteny analyses of genes encoding the pore-forming α subunit of $Na_V$ lead to different conclusions regarding *scn4a* (S9 Fig). Two rounds of whole-genome duplication (WGD) have occurred at the base of the vertebrate lineages [29,30]. The extant *scna* gene family in vertebrates comes from a single *scna* gene that went through these two rounds of WGD. An additional third round of WGD specifically happened in an ancestor of the teleosts [31,32]. In humans and mice, the *SCNA* gene family includes 10 members that come from 4 *scna* genes resulting from the second round of WGD. Two went through additional series of single gene duplications, mostly in tandem, leading to two clusters of *SCNA* genes, whereas *SCN4A* and *SCN8A* experienced no further duplication. In zebrafish and other teleostean fish, the third WGD led to 8 *scna* genes with 2 *scn4a* genes. Interestingly, *scn4aa* and *scn4ab* lie on pairs of chromosomes that derived from the teleost-specific third WGD (S9 Fig) [31,33,34]. These syntenic and phylogenetic data, including protein identity data, strongly suggest that *scn4aa* and *scn4ab*, which are quasi-exclusively expressed in skeletal muscles, have been conserved during evolution because they are required for muscle function in zebrafish [10,11]. At first sight, these data are poorly consistent with our findings on zebrafish $Na_V$ properties and the absence of phenotype in the digenic *scn4aa* and *scn4ab* knock-out mutant line. However, we found that $Na_V$ density estimated by maximal $Na^+$ conductance per cell capacitance was not significantly different between zebrafish and extra-synaptic mouse muscle sarcolemma. Like phylogenetic data, the maintenance of $Na_V$ density in teleosts argues in favor of a significant role played by these channels in zebrafish muscle function. In that context, if the primary role of $Na_V$ in zebrafish muscle is presumably not to generate and propagate APs, a putative role of $Na^+$ channels may be that activation of inward depolarizing currents through the 20% of $Na_V$ that are not in an inactivated state at −80 mV can add to the endplate inward depolarizing current to reach higher voltages, constituting in this way a safety factor for activating maximal SR $Ca^{2+}$ release and force production.

## Materials and methods

### Ethics statement

All animal manipulations were performed in agreement with the European Directive on using laboratory animals (2010/63/EU), the ethics principles of the French Department of Veterinary Services and the French Ministry for Higher Education,

Research and Innovation. Manipulations of mice were done in accordance with the guidelines of the local animal ethics committee at University Claude Bernard Lyon 1 (Animal Experimentation Committee no. CEEA-015), the French Ministry of Agriculture (decree 87/848). Manipulations of zebrafish mutant lines were done in accordance with the guidelines of the local animal ethics committee SBEA RAM-iExplore (Animal Experimentation Committee no. CEEA-LR36) and conducted following local (authorization #3807) and national (authorization #24696) ethic approvals.

## Animals

Adult fish were reared under standardized conditions (28 ± 0.5 °C, pH 7, 14/10 h light/dark photoperiod) in a recirculation system (Techniplast) and fed twice or third a day with dry pellets (Gemma, Skretting). For electrophysiological analyses, WT zebrafish (mixed AB/TU genetic background) were used. Males (11–13-week-old) were euthanized with anesthetic overdose (tricaine MS222 solution, 200 mg/l) and decapitated. Skin was removed and deep layers of muscles located in the dorso-caudal region were dissected. Samples were then incubated at 37 °C in a Tyrode solution containing collagenase type 1 (1 mg/ml, Sigma-Aldrich) for 40 min. After enzyme treatment, muscles were rinsed and stored in Tyrode solution at 5 °C until use. Muscle fibers were then isolated as described below.

Enu-induced mutant alleles for *scn4aa* and *scn4ab* genes were selected using the zebrafish information network (ZFIN http://zfin.org/) database [12]. Eggs with the mutant alleles were obtained from Zebrafish International or European Zebrafish Resource Centres (ZIRC and EZRC, respectively). Monogenic (one mutant allele in *scn4aa* or *scn4ab*) and digenic (one mutant allele in *scn4aa* and one mutant allele in *scn4ab*) mutant lines were established on AB genetic background by outcross with WT animals over a minimum of 5 generations. Digenic *scn4aa*$^{-/-}$;*scn4ab*$^{-/-}$ homozygous mutant and *scn4aa*$^{+/+}$;*scn4ab*$^{+/+}$ homozygous control animals were obtained by intercrossing heterozygous *scn4aa*$^{+/-}$;*scn4ab*$^{+/-}$ fish.

Adult male OF1 mice (8–16-month) were killed by cervical dislocation before the removal of *flexor digitorum brevis* muscles. Muscles were incubated in a Tyrode solution containing 2 mg/ml collagenase type I (Sigma-Aldrich) for 50 min at 37 °C. As for zebrafish muscles, enzymatically treated muscles were then rinsed in Tyrode solution and muscle fibers were next isolated as described below.

## Electrophysiology

**Whole-cell voltage- and current-clamp recordings.** Single intact muscle fibers were separated from zebrafish or mouse muscles by gentle mechanical trituration of the collagenase-treated muscles in a glass-bottomed experimental chamber, in the presence of IMDM culture medium (Gibco). For zebrafish muscles, trituration was performed in IMDM previously supplemented with 2% matrigel (Sigma-Aldrich). Moreover, isolated zebrafish muscle fibers were exclusively fast type, easily identifiable on the basis of their morphology and their ability to develop Na$_V$ currents and APs [35].

Prior to trituration, the bottom of the experimental chamber was covered with a thin layer of silicone grease. This enabled single fibers from zebrafish or mice to be covered with additional silicone so that a 50–100 µm-long portion of the fiber extremity was left out, as previously described [35]. The culture medium solution was replaced by the extracellular solutions (see Solutions). The tip of a glass micropipette filled with an intracellular-like solution (see Solutions) was inserted into the silicone-embedded fiber portion. The tip of the micropipette was gently crushed against the bottom of the chamber inside the fiber to reduce the electrode resistance to a few hundreds of kΩ and to allow internal dialysis of the fiber. The silver-silver chloride wire inside the micropipette was connected to an RK-400 patch-clamp amplifier (Bio-Logic, Claix, France) used in whole-cell voltage- or current-clamp configuration. Command voltage or current pulse generation was achieved with an analog-digital converter (Digidata 1400A, Molecular Devices, San Jose, CA, USA) controlled by pClamp 10 software (Molecular Devices). Analog compensation was systematically used to decrease the effective series resistance. Currents in voltage clamp experiments and voltage changes in current clamp experiments were acquired at a sampling frequency of 50 kHz. In voltage-clamp experiments, cell capacitance was determined by integration of a current

trace obtained with a 10-mV hyperpolarizing pulse from the holding potential preceding each voltage step and was used to calculate the density of currents (A/F). Leak currents were subtracted from all recordings using the same pulse preceding every test pulse supposing a linear evolution of leak current with depolarization. After leak subtraction, the $Na^+$ current amplitude was determined by subtracting the current amplitude at the end of the voltage step to the current amplitude at peak. At the beginning of current clamp experiments, positive 0.5-ms current steps of increasing amplitudes were applied to determine the threshold for AP activation. This current was re-adjusted upon lowering resting membrane potential. All experiments were performed at room temperature (20–22 °C).

**Solutions.** The dialyzed intracellular solution contained (in mM) 140 K-glutamate, 5 $Na_2$-ATP, 5 $Na_2$-phosphocreatine, 5.5 $MgCl_2$, 5 glucose, and 5 HEPES, adjusted to pH 7.2 with KOH. The external solution used for recording voltage-gated $Na^+$ current (Figs 1B and 2) contained (in mM) 120 NaOH, 20 tetraethylammonium-OH (TEA), 1 4-aminopyridine, 1 $MgCl_2$, 2.5 $CaCl_2$, 10 HEPES, adjusted to pH 7.2 with methanesulfonic acid ($MeSO_3^-$). The external Tyrode solution for recording APs and the currents presented in Fig 1A and 1B (Tyrode) contained (in mM) 140 NaCl, 5 KCl, 1 $MgCl_2$, 2.5 $CaCl_2$, 10 HEPES, adjusted to pH 7.2 with NaOH. Stock solution of TTX (Sigma) was prepared in water at 100 μM and stored at −20°C. In Fig 1B, the fiber was exposed to different solutions by placing it in the mouth of a perfusion tube from which the rapidly exchanged solutions flowed.

## Microscopic analysis of motor endplates of zebrafish muscle fibers

**Labeling of zebrafish muscle fibers.** Isolated skeletal muscle fibers or small bundles of fibers were separated from the collagenase-treated muscle mass by gently flushing Tyrode solution onto the muscle. Isolated or bundled fibers were allowed to sediment in a 1 ml microtube. Tyrode was then discarded and replaced by fixative (4% paraformaldehyde in phosphate-buffered saline [PBS] 1×) for 45 min. After successive washes with PBS 1× (3 times, 10 min), labeling of AChRs and nuclei was performed using respectively TRITC-coupled α-bungarotoxin (1:100; ThermoFisher scientific) and Hoechst (1:250,000; ThermoFisher Scientific) diluted in PBS 1× (overnight, 4 °C). Washes with PBS 1× were then performed (3 times, 10 min). Stained fibers were deposited on a glass slide and intact fibers were gently separated from each other under a binocular magnifier using thin forceps. Samples were mounted with a glass coverslip and Prolong glass antifade mounting medium (Invitrogen).

**Microscopy imaging and analysis.** Zeiss apotome 3 microscope system and 20× objective were used to capture Z-stacks images encompassing the entire volume of muscle fibers by means of 0.49 μm-thick optical slices. Distance analysis of AChRs labeling was then performed using ImageJ software and the Euclidean distance map function. Image stacks of α-bungarotoxin labeled skeletal muscle fibers were first straightened in order to align main axis of fibers with main axis of the images. Measurements were made on upper and lower sarcolemma by dividing each fiber at its equator and applying Z-projection to each half. After thresholding, images were processed using Euclidean distance transformation. The Euclidean distance map represents the shortest distance to the nearest AChRs clusters, *i.e.,* labeled pixels, for each nonlabeled pixel of the extra-synaptic sarcolemma. Corresponding data were collected on Z-stack images of 10 α-bungarotoxin-labeled fibers (3 WT and 4 mutant fish, each from independent experiments).

## Molecular analyses

**Mutant line genotyping.** Genotyping 4-month-old animals was done on fin biopsies. DNA was extracted using hot sodium hydroxide and Tris-HCl [36]. Amplicons containing the point mutations were obtained by PCR amplification (One Taq polymerase, New England Biolabs) using the following primers: sa1270 forward, 5′-CTTTTTCGCATCACCCTT-3′ and sa1270 reverse, 5′-GTAGGTGTCAAAGCTACT-3′ (annealing temperature of 58 °C, amplicon of 437 bp); sa33152 forward, 5′-TGCAGCAGGAAAGACGTACA-3′ and sa33152 reverse, 5′-AAAGCACTCTCTACAGCGCT-3′ (touch-down PCR from 65 to 55 °C, amplicon of 492 pb). PCR products were enzymatically purified (Thermosensitive FastAP, Thermo Scientific) and Sanger sequenced (Azenta Life Sciences). Sequences were analyzed using SnapGene Viewer.

**mRNA analyses.** Skeletal muscles (trunk and tail) were dissected from adult animals euthanized in tricaine solution (240 mg/L) and used for total RNA extraction with MaXtrac-high density (Qiagen) and TRIzol reagent (Thermo Scientific) after mechanical homogenization (TissuRuptor II, Qiagen). Total RNA was cleaned with RNA cleanup XS column (Macherey-Nagel), and 2 µg was reverse transcribed using SuperScript IV reverse transcriptase and random hexamers (Thermo Scientific) after digestion with deoxyribonuclease I, amplification grade (Thermo Scientific). First strand cDNAs were amplified by touch-down PCR (from 65 to 55 °C, One Taq polymerase, New England Biolab) with the following primers: *Scn4aa* exons 9–13 (5′-GGCTCGTGACAAAGAAGAGG-3′ and 5′-GTTAAGGGTGGGCCAAGATTT-3′), *Scn4ab* exons 8–12 (5′-GGCTACACCAGTTATGACAAC-3′ and 5′-TAATATGGGTCCAGCGCA-3′) and eef1a1l1 as housekeeping gene (5′-CTTCTCAGGCTGACTGTGC-3′ and 5′-CCGCTAGCATTACCCTCC-3′). PCR products were evaluated on 2% agarose gels and sequenced as done for the genotyping procedure using the PCR primers.

Real-time PCR analyses of *scn4aa* and *scn4ab* transcripts were done on skeletal muscles from adult fish using specific PrimePCR primers designed on demand (Bio-Rad), and SYBR green chemistry with LightCycler 480 SYBR green I master (Roche Life Science). The primer sequences for the eef1a1l1 referent gene are 5′ATCAACTGCGTGGTCGAAG-3′ and 5-TGAAGGCAGCAATATCCACA-3′. Analyses were done with the LightCycler software (1.5.1.62).

**Protein alignment.** The referent sequences were hsNa$_v$1.4a, P35499.4; mmNa$_v$1.4a, NP_573462.2; drNa$_v$1.4aa, Q2XVR3.2; drNa$_v$1.4ab, Q20JQ7.1. The "align two or more sequences" option of the Blastp suite (National Center for Biotechnology Information) and the Constraint-based Multiple Alignment Tool (COBALT, conservation setting: 3 bits) were used to align sequences and determine the identity between proteins.

## Swimming analyses

Videos of the free swimming and escape response of adult zebrafish were recorded with a Xiaomi camera (Redmi Note 13 Pro+) at 60 frames/s (free swimming, 10 s long videos) and 240 frames/s (escape response, 1 s long videos). Two similar objects (1 g) placed at approximately 30 cm above the water were simultaneously dropped into two adjacent tanks (1.6 L) containing controls or mutants (3 or 4 adult fish by tank) to induce escape responses. Adult fish (4–12-months of age, both sex) were studied. Body length of fish were similar between controls and mutants at similar ages (2.6 ± 0.2 cm for 4-month-old and 3.6 ± 0.2 for 12-month-old fish). Videos were processed with Fiji software (angle tool and manual tracking plugin). For escape response, minimal head-body angle, maximal and mean speed, and event latencies were calculated for each responding fish within the 40 ms from the initial response. Graphs were drawn with GraphPad prism 10.

## Statistics

Statistical analysis was performed using Microcal Origin, GraphPad Prism, and R. Least-square fits were performed using a Marquardt–Levenberg algorithm routine included in Microcal Origin. Data are given as means ± SEM in graphs (Fig 2B and 2C) and as means.. *t* test taking into account the number of fibers from each animal as described in Eisner [37]. Unpaired Mann–Whitney tests were used for swim analyses of adult fish. Wilcoxon nonparametric test was used for statistical analysis of mean and maximal distances to AChR clusters. Numbers of individual measurements and individual animals used are mentioned in the figure legends or in the main text. Differences were considered significant when $P < 0.05$, and *, **, and *** indicate $P < 0.05$, $P < 0.005$, and $P < 0.0005$, respectively.

## Supporting information

**S1 Fig. Resting membrane potential measured in trunk muscle fibers using an intracellular microelectrode.**
Resting membrane potential was measured in situ in skeletal muscle fibers in euthanized WT zebrafish using intracellular microelectrodes and a micro-electrode amplifier (VF 1800, BioLogic, France) in the presence of an extracellular Tyrode solution containing 2 mM K$^+$ and an intramicroelectrode medium containing 3 M KCl. The microelectrodes resistance was

between 20 and 30 MΩ. Measurements were done on muscle fibers located in the deep part of the trunk accessible to microelectrodes from the interior of the fish after having removed the internal organs and a thin film lining the interior of the thorax. All the fibers are fast-twitch type in this trunk region. Three different fibers were impaled with the same microelectrode that was slightly displaced laterally between each impalement. The mean resting membrane potential was found to be −78±0.8 mV in 124 fibers from 4 fish. The data underlying S1 Fig can be found in S1 Data.
(PDF)

**S2 Fig. Effect of increasing voltage on Ca$^{2+}$ signals elicited by field stimulation in an isolated WT zebrafish fast skeletal muscle fiber.** Fibers were loaded with the acetoxymethyl ester form of the Ca$^{2+}$ fluorophore indo-1 (2 µM; Invitrogen) during 30 min and then subsequently washed (30 min) in Tyrode solution to allow de-esterification of the dye. Indo-1 fluorescence was measured on an inverted Nikon Diaphot epifluorescence microscope equipped with a commercial optical system, allowing the detection of fluorescence at 485 nm by a photomultiplier (IonOptix, Milton, MA, USA) upon 360 nm excitation. Fluorescence signals were acquired at a sampling frequency of 10 kHz. Fibers were electrically stimulated through parallel platinum field electrodes at 1 Hz with pulses of 0.5 ms duration. The voltage was gradually increased from 8 to 30 V, as indicated by the numbers below the lower trace. Arrows point to Ca$^{2+}$ signals presented on an expanded scale. The stimulation frequency was 1 Hz, and the pulses duration was 0.5 ms. Graded responses to increasing voltages were observed in all the 8 tested fibers from 2 WT fish.
(PDF)

**S3 Fig. Absence of effect of tetrodotoxin (TTX) (2 µM) on Ca$^{2+}$ signals evoked by field stimulation in an isolated WT zebrafish fast skeletal muscle fiber.** Changes in indo-1 fluorescence were measured using the same procedures as the ones indicated in the legend of S2 Fig in the absence (left) and after addition (right) of 2 µM TTX in the same fiber. Fibers were electrically stimulated at 0.4 Hz with pulses of 30 V amplitude and 0.5 ms duration. Absence of effect of TTX was observed in all the 10 fibers from 3 WT fish tested with this protocol.
(PDF)

**S4 Fig. Effect of tetrodotoxin (TTX) (2 µM) on Ca$^{2+}$ signals evoked by field stimulation in an isolated mouse skeletal muscle fiber.** Changes in indo-1 fluorescence were measured using the same procedures as the ones indicated in the legend of S2 Fig in the absence (left), in the presence (middle), and after washout (right) of 2 µM TTX in the same fiber. The stimulation frequency was 0.25 Hz, and the pulses duration was 0.5 ms. Note that all-or-none Ca$^{2+}$ signals were evoked in response to pulses of lower voltage (4 V) as compared to zebrafish. The arrow points to Ca$^{2+}$ signal presented on an expanded scale. An inhibitory effect of TTX was observed in all the 10 tested fibers from 2 mice.
(PDF)

**S5 Fig. Analysis of gene expression for *scna* genes.** PCR amplification was done using cDNA samples obtained by reverse-transcription of total RNA samples extracted from trunk muscles of control (C) and digenic *scn4aa*$^{-/-}$;*scn4ab*$^{-/-}$ mutant (DM) adult fish (3 fish by genotype) and control (L+) and digenic mutant (L−) 6-days-post fertilization(dpf) larvae (pool of 30 larvae). Genomic DNA (D, 25 ng) was used to demonstrate the absence of genomic DNA contamination in cDNA samples (no PCR products or PCR products of higher size), and water (0) to demonstrate the absence of any PCR contamination. The *eef1a1l1* gene was used as control of reverse-transcription efficiency. Primers used are listed in S2 Table. A touch-down protocol was used for PCR amplification. A single amplicon at the expected size was obtained for all genes in the larval samples as expected, since whole larvae were used for RNA extraction. In trunk muscle extracts of control and double mutant samples, PCR amplicons were only obtained for *scn4aa* and *scn4ab*, confirming that *scn4a* are the sole *scna* isoforms expressed in control muscles of zebrafish and suggesting the lack of upregulation of another *scna* gene expression in muscles of double mutant fish. Created in BioRender. NICOLE, S. (2025) https://BioRender.com/ddf4c4j.
(PDF)

**S6 Fig. Alignment of human (hsNav1.4a), mouse (mmNa$_V$1.4a), and zebrafish (drNa$_V$1.4aa and drNav1.4ab) Na$_V$1.4a proteins.** Amino acids conserved between mouse and zebrafish proteins are in red. The 6 transmembrane segments (S) of the 4 domains (D) are underlined in yellow, the pore-forming loops are underlined in gray, and the inactivation gate is underlined in green.
(PDF)

**S7 Fig. *scn4a* point mutations studied and their impact on mutant muscle mRNAs. (A)** Schematic representation for Na$_V$1.4a inserted into the membrane with colors indicating the degree of identity between mammalian and zebrafish Na$_V$1.4a proteins. The position of the *scn4aa* (Na$_V$1.4aa) and *scn4ab* (Na$_V$1.4ab) mutations is indicated. **(B)** Effect of the c.T1498-1g>a splicing mutation on *scn4aa* mRNA. **(a)** This point mutation is located in the acceptor splice site of intron 10 where it substitutes the invariable G nucleotide of the highly conserved AG sequence by an A. The PCR primers used to investigate its effect on *scn4aa* mRNA splicing are located in exons 9 and 13 (arrows). A single amplicon at the expected size (793 pb) was obtained when amplifying *scn4aa* cDNAs from adult skeletal muscles of zebrafish homozygous for the mutation (MT), similar to control animals from the same lays (CT). Left well: 100 pb ladder. **(b)** Sanger sequencing electropherograms of the PCR amplicons obtained from CT and MT muscle samples showed that the mutant *scn4aa* cDNA is deleted for 10 nucleotides (in bold) in the 5′ region of exon 11 (red horizontal bar and vertical arrow in the CT and MT electropherograms, respectively). A cryptic acceptor splice site (AG motif in the 3′ region of the deleted sequence) is used for intron 10 splicing in mutant *scn4aa* mRNA. (C) Effect of the c.1393A>T nonsense mutation on *scn4ab* mRNA. **(a)** A single amplicon at the expected size (800 pb) was obtained when amplifying *scn4ab* cDNAs from adult skeletal muscles of zebrafish homozygous for the mutation (MT) with PCR primers located in exons 8 and 12 (arrows). Left well: 100 pb ladder. **(b)** Sanger sequencing electropherograms of the PCR amplicons obtained from homozygous CT and MT muscle samples showed that all the mutant *scn4ab* cDNAs contain the nonsense mutation. **(D)** Quantitative real-time PCR analyses of *scn4aa* and *scn4ab* gene expression in trunk muscles of control and mutant adult fish did not argue for a decreased amount of *scn4a* mRNAs in double mutant (DM) samples compared to control (CT) samples using *eef1a1l1* mRNA as a reference. Three independent fish by genotype were used (squares). The qRT-PCR experiment was done in triplicate by sample on 2 independent reverse-transcribed preparations. Gel images in (B.a) and (C.a) were done by splicing 2 (B.a) and 3 (C.a) fragments of the same original image, pointed by vertical black lines, in order to remove irrelevant lanes. Created in BioRender. NICOLE, S. (2025) https://BioRender.com/0bg9p10. The data underlying S7D Fig can be found in S1 Data.
(PDF)

**S8 Fig. Quantitative analyses of free swimming in control and mutant fish. (A)** Total distance covered in 10 s. **(B)** Maximal speed. Graphs represent mean ± SD with individual values (unpaired Mann–Whitney test) (4 to 12-month-old, minimum of 10 fish per genotype). The data underlying S8A and S8B Fig can be found in S1 Data.
(PDF)

**S9 Fig. *SCN* gene family amplification in the course of vertebrate evolution.** All the *SCNA* genes observed in vertebrates derive from a single *SCNA* gene of their latest common ancestor (gray). This gene went first through 2 rounds of whole-genome duplications (WGDs) to lead to 4 *SCNA* genes in the early vertebrate genome. The *SCNA* gene repertoire amplified furthermore, but in different modes of duplications according to genome of fish and mammalian lineages. A third WGD specific to the teleost fish lineage leads to 8 *scna* genes in those fish, while in human, tandem single gene duplications led to two clusters: one composed of *SCN5A*, *SCN10A*, *SCN11A*, and a second cluster containing *SCN3A*, *SCN2A*, *SCN1A*, *SCN9A*, and *SCN7A*. *SCN4A* and *SCN8A* do not show any further tandem duplications. Dr: *Danio rerio* (zebrafish; Cypriniformes, *Ostariophysii*); and three Percomorpha (*Neoteleostei*) OI: *Oryzias latipes* (medaka); Ga: *Gasterosteus aculeatus* (stickleback); and Tn: *Tetraodon nigroviridis* (green spotted puffer). Chromosome numbers (or scaffold for Ga) are provided, unless not available (NA) or unassigned (un). In the human genome, the two clusters found on chromosome

2 are separated by three genes (gray boxes); Arrows represent the gene orientation (as depicted in Genomicus [38]). (Free silhouette images of organisms are from PhyloPic, version 2.0 (https://www.phylopic.org/nodes)).
(PDF)

**S1 Video. Video recording of freely swimming in control and mutant zebrafish.** The tank containing mutant fish is on the top of the video (60 fps, 10 s).
(MP4)

**S2 Video. Video recording of escape response to an aversive stimulus in control and mutant zebrafish.** The tank containing mutant fish is on the top (240 fps, 1 s).
(MP4)

**S1 Table. Percent Identity Matrix between human, mouse, and zebrafish Na$_V$1.4a proteins.**
(DOCX)

**S2 Table. Primers used for analyzing gene expression by PCR.**
(DOCX)

**S1 Raw Images. Raw images for gels in S5 and S7 Figs.**
(PDF)

**S1 Data. Numerical data of S1, 2D, 3B, 4C, S7D and S8B Figs.**
(XLSX)

## Acknowledgments

We acknowledge the contribution of the staff of the ANIPHY facility, the Animalerie Zebrafish Rockefeller (AZR) (SFR Santé Lyon-Est, UCBL, UAR3453/CNRS, US7/Inserm) and the Zefix ZebKO (BioCampus Montpellier) platform, especially Aurélien Drouard, for their help in mice and zebrafish care and maintenance, respectively. We also acknowledge the contributions of the CELPHEDIA Infrastructure (http://www.celphedia.eu/), especially the center AniRA in Lyon, Youssef Issa and Clémence Bronstein for their help.

## Author contributions

**Conceptualization:** Sophie Nicole, Bruno Allard.

**Formal analysis:** Christine Berthier, Léon Espinosa, Sophie Nicole, Bruno Allard.

**Funding acquisition:** Vincent Jacquemond, Philippe Lory, Sophie Nicole, Bruno Allard.

**Investigation:** Léa Demesmay, Romane Idoux, Claire Bernat, Angel Maunier-Mercier, Sophie Nicole, Bruno Allard.

**Methodology:** Romane Idoux, Claire Bernat, Léon Espinosa, Sophie Nicole, Bruno Allard.

**Project administration:** Sophie Nicole, Bruno Allard.

**Software:** Frédéric Brunet.

**Supervision:** Sophie Nicole, Bruno Allard.

**Validation:** Sophie Nicole, Bruno Allard.

**Visualization:** Sophie Nicole, Bruno Allard.

**Writing – original draft:** Christine Berthier, Sophie Nicole, Bruno Allard.

**Writing – review & editing:** Christine Berthier, Vincent Jacquemond, Frédéric Brunet, Philippe Lory, Sophie Nicole, Bruno Allard.

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
