## [Editor Report · Decision Letter 0]

10 Jan 2025

Dear Dr Allard,

Thank you for submitting your manuscript entitled "Challenging a dogma: uselessness of voltage-gated Na+ channels for fast muscle fibers of zebrafish to twitch" for consideration as a Research Article by PLOS Biology.

Your manuscript has now been evaluated by the PLOS Biology editorial staff and I am writing to let you know that we would like to send your submission out for external peer review.

Also, we would like to consider your manuscript as a Short Report. Please select this article format when you complete the full submission. While it is not necessary at this stage to reformat the manuscript, if the manuscript progresses further toward publication we ask that you combine data elements so that there are four or fewer figures in total.

Once your full submission is complete, your paper will undergo a series of checks in preparation for peer review. After your manuscript has passed the checks it will be sent out for review. To provide the metadata for your submission, please Login to Editorial Manager (https://www.editorialmanager.com/pbiology) within two working days, i.e. by Jan 12 2025 11:59PM.

Kind regards,

Taylor

Taylor Hart, PhD,

Associate Editor

PLOS Biology

thart@plos.org

---

## [Decision Letter · Decision Letter 1]

18 Feb 2025

Dear Dr Allard,

Thank you for your patience while your manuscript "Challenging a dogma: uselessness of voltage-gated Na+ channels for fast muscle fibers of zebrafish to twitch" was peer-reviewed at PLOS Biology. Your manuscript has been evaluated by the PLOS Biology editors, an Academic Editor with relevant expertise, and by several independent reviewers.

As you will see in the reviewer reports, which can be found at the end of this email, although the reviewers find the work potentially interesting, they have also raised a substantial number of important concerns. Based on their specific comments and following discussion with the Academic Editor, it is clear that a substantial amount of work would be required to meet the criteria for publication in PLOS Biology. However, given our and the reviewer interest in your study, we would be open to inviting a comprehensive revision of the study that thoroughly addresses all the reviewers' comments. Given the extent of revision that would be needed, we cannot make a decision about publication until we have seen the revised manuscript and your response to the reviewers' comments. Your revised manuscript would need to be seen by the reviewers again, but please note that we would not engage them unless their main concerns have been addressed.

You’ll see that the reviewers think that the study reports compelling data and would be of interest. However, none of them are convinced by the authors' characterization of the NaV channel mutant zebrafish. All reviewers recommended electrophysiological characterization of the mutant fish muscles as well as more thorough behavioral analysis. R1 and R2 pointed out the need to test for developmental compensation. R2 pointed out the need to confirm loss of functional NaV channels from the muscles. R3 also requested more careful e-phys characterization in the wild type fish muscles.

Having discussed the reviews with the Academic Editor, we emphasize the need to perform new experiments in the wild type and mutant fish. These should include electrophysiological studies describing the resting potentials and capacity for APs in the muscles, as well as characterizing the tissue to test for compensation mechanisms in the mutants. We would also strongly suggest that you carry out more thorough behavioral study of the mutants.

IMPORTANT: For this round of review, we asked the reviewers to consider your manuscript as a Short Report. However, because this format allows a maximum of 4 Figures, and you currently have 7 Figures, you will need to reduce the number of Figures, either by combining them, or by moving some material to the supplement. We do concede that it is possible that your revisions might make this a much more substantial piece of work, in which case we could consider reverting to a full Research Article.

We appreciate that these requests represent a great deal of extra work, and we are willing to relax our standard revision time to allow you 6 months to revise your study. Please email us (plosbiology@plos.org) if you have any questions or concerns, or envision needing a (short) extension.

**IMPORTANT - SUBMITTING YOUR REVISION**

*Resubmission Checklist*

*Published Peer Review*

*PLOS Data Policy*

*Blot and Gel Data Policy*

Sincerely,

Taylor

Taylor Hart, PhD,

Associate Editor

PLOS Biology

thart@plos.org

REVIEWS:

Reviewer #1: This manuscript has been submitted to PLoS Biology because it claims to challenge the dogma about action potentials being required for movement. The authors have previously shown that adult zebrafish muscle fibers show superfact "kinetics of action potentials" and excitation-contraction coupling. In this manuscript the authors question whether action potentials are necessary for movement. This conclusion appears to be based on a few different threads of evidence. One is that the resting potential of zebrafish muscle fibers is lower than mouse muscle fibers (although aren't resting potentials usually different between different cells?). It could be more informative to perturb the system and show that the two species of muscle fibers respond differently. The authors note that a sodium channel blocker does not block movement in zebrafish (trichaine). The authors show that acetylcholine receptors are continuously distributed along the adult muscle fiber, leading the authors to hypothesize that action potential propogation may not be necessary for fiber depolarization and contraction. The authors then test this hypothesis by generating double mutants for two sodium channel genes scn4aa and acn 4ab and these mutants move normally.

It is this reviewers opinion that the data shown to not surpass the bar for a PLoS Biology report claiming that sodium channels/action potentials are not necessary. The data are certainly compelling, but the burden of proof for a negative result has not been met. For example, although the authors say there are two sodium channels in zebrafish it would be more correct to say there are two alpha subunits - and 14 total. Without an understanding of the impacts of the other subunits on potential compensatory activity, without understanding the impact of the double mutants on muscle physiology and muscle structure, it seems like it is premature to claim that these are not necessary. Is it at all possible that muscle fibers incorporate more slow characteristics, and perhaps acceleration is impacted whereas the average speed is not? The data shown are all in the first 32ms following the stimulus. Why was this time frame chosen? What about normal motility tests that analyze dark/light average swim speeds/velocity/acceleration over longer time frames? Is it possible that the escape response does not need these subunits, and other subunits/muscle types can compensate during general swimming? Is it known in other systems what the impact of deleting the alpha subunit is? How is resting potential altered? How is Ca2+ signaling altered? Is there increased innervation, are there more AChR receptors per fiber in the double mutants? These questions are ones that could help the authors present a more compelling story.

-----

Reviewer #2: This study challenges the established role of voltage-gated sodium channels (NaV) in zebrafish fast muscle fibers by presenting electrophysiological and genetic evidence suggesting that these channels are not essential for muscle contraction. The topic is interesting and has implications for our understanding of muscle physiology and neuromuscular diseases. However, while the study presents robust electrophysiological data, several significant weaknesses limit its impact. In particular, the analysis of NaV mutants is incomplete and lacks crucial complementary approaches such as immunohistochemistry and further electrophysiological characterization. Below, I detail my concerns and suggest necessary revisions to improve the manuscript.

The study generates double knock-out (scn4aa-/-, scn4ab-/-) zebrafish but does not provide adequate validation of the loss of NaV channels at the protein level. The absence of functional NaV in these mutants is inferred but not directly demonstrated. Immunohistochemistry should be performed to confirm the complete loss of NaV protein in muscle fibers.Western blot analysis or quantitative PCR showing mRNA non sense mediated decay could further substantiate the absence of NaV expression in skeletal muscle.

Skeletal muscle could have plasticity in response to genetic perturbations. The study does not examine whether alternative mechanisms compensate for the loss of NaV, such as changes in ion channel expression or synaptic architecture. A transcriptomic or proteomic analysis would help determine whether other ion channels are upregulated in response to NaV loss. Electrophysiological studies of synaptic inputs, such as miniature endplate potentials (mEPPs) or spontaneous synaptic currents, would provide insights into neuromuscular transmission in mutants.

The authors claim that NaV deletion does not affect locomotion but provide only qualitative observations. More quantitative locomotion assays, such as high-speed videography with kinematic analysis, should be conducted.

Overall, the conclusion that zebrafish fast muscle fibers do not require NaV for contraction is intriguing but may be overstated. A possibility not considered is that NaV loss might lead to developmental compensation that is not present in wild-type fish. An analysis of neuromuscular junction development by immunohistochemisty in mutant and WT should also be performed.

-----

Reviewer #3: In this manuscript the authors present data to argue that skeletal muscle voltage gated Na channels (NaV1.4) do not perform the typical function associated with these channels in adult zebrafish fast twitch skeletal muscle fibers. The authors use a variety of techniques including electrophysiology, imaging, molecular and behavioral tools to show that adult zebrafish fast twitch fibers produce APs that peak at a lower voltage than mouse fast twitch fibers, are shifted towards more negative voltages in their activation and inactivation kinetics, and cannot elicit robust APs at resting potentials more depolarized than around -80 mV. The authors find that synaptic terminals and endplate contacts are located in close proximity and that genetic manipulation leading to inactive or non-functional NaV1.4 channels does not impact swimming in adult fish.

The work is well done and provides a reasonable argument to support the author's conclusions. The comparison with mouse fibers is compelling and is an important aspect of the study. It provides a control for the techniques and suggests that the experimental protocols are well founded. I have a few comments below but my main concern relates to confirming the resting potential of these fibers in adult animals (or as close to in situ as possible), and voltage and current clamping mutant fibers as mentioned below.

Comments are listed below.

Comments:

1. Results, Pg 5: I am surprised that the voltage threshold for Na+ current activation is significantly different between the mouse (-65 +/- 6 mV) and zebrafish (-59 +/- 7 mV). Please double check.

2. Results, Pg 6: A determination of the resting membrane potential in adult zebrafish fast twitch fibers would be beneficial to the study. The Westerfield work is very good, but when presenting such a provocative finding as the authors have done, it would be ideal to record the data under conditions in which as little manipulation is performed as possible. In other words, what is the true resting membrane potential of these fibers in situ?

3. I assume these are fast twitch fibers. If so, do they still twitch and contract when Nav1.4 channels are inactive/non-functional? Do the authors know if twitching occurs in mutants?

4. I am surprised that the authors did not attempt to record APs from mutants to show physiologically that a) these are indeed the only NaV channels that are present and functional, and b) that their mutations did indeed result in non-functional channels. I feel that these should be done.

5. Discussion, Pg 11: The sentence in lines 8-13 contains repetitive text and should be reviewed.

---

## [Decision Letter · Decision Letter 2]

3 Oct 2025

Dear Dr Allard,

Thank you for your patience while we considered your revised manuscript "Challenging a dogma: uselessness of voltage-gated Na+ channels for fast muscle fibers of adult zebrafish to twitch" for publication as a Short Report at PLOS Biology. This revised version of your manuscript has been evaluated by the PLOS Biology editors, the Academic Editor and two of the original reviewers.

Based on the reviews, we are likely to accept this manuscript for publication. Please also make sure to address the following data and other policy-related requests.

IMPORTANT: Please ensure that your next revision addresses the following points to avoid delays in publication:

----------

**Title:

We would like to modify your title for clarity and to match our stylistic preferences. We suggest the following alternative version:

"Zebrafish fast muscle contractions avoid the mammalian requirement for voltage-gated Na+ channels"

**Financial disclosure statement:

-- Please add links to the funding agencies in the Financial Disclosure statement in the manuscript details.

**Data:

-- For ease of reproducibility, we request some additional items related to your data provision.

--Please supply the numerical values either in a supplementary excel file or as a permanent DOI’d deposition for the following figures:

2D

3BEFG

4C

S7D

S8AB

-- Please cite the location of the data clearly in all relevant main and supplementary Figure legends, e.g. “The data underlying this Figure can be found in S1 Data” or “The data underlying this Figure can be found in https://doi.org/10.5281/zenodo.XXXXX”

-- Please ensure that you are using best practice for statistical reporting and data presentation. These are our guidelines https://journals.plos.org/plosbiology/s/best-practices-in-research-reporting#loc-statistical-reporting and a useful resource on data presentation https://journals.plos.org/plosbiology/article?id=10.1371/journal.pbio.1002128

-- If you are reporting experiments where n ≤ 5, please plot each individual data point.

-- Supplementary files (e.g., excel). Please ensure that all data files are uploaded as 'Supporting Information' and are invariably referred to (in the manuscript, figure legends, and the Description field when uploading your files) using the following format verbatim: S1 Data, S2 Data, etc. Multiple panels of a single or even several figures can be included as multiple sheets in one excel file that is saved using exactly the following convention: S1_Data.xlsx (using an underscore).

-- Please ensure that your Data Statement in the submission system accurately describes where your data can be found and is in final format, as it will be published as written there.

**Gels:

-- Please ensure that you provide the original, uncropped and minimally adjusted images supporting all blot and gel results reported in the Figures.

-- We will require these files before a manuscript can be accepted so please prepare and upload them now. Please carefully read our guidelines for how to prepare and upload this data: https://journals.plos.org/plosbiology/s/figures#loc-blot-and-gel-reporting-requirements

----------

We expect to receive your revised manuscript within one week.

*Published Peer Review History*

*Press*

Sincerely,

Taylor

Taylor Hart, PhD,

Associate Editor

thart@plos.org

PLOS Biology

REVIEWS

Reviewer #1 [Clarissa Ann Henry]: The authors have worked to address reviewer comments. This manuscript is interesting and the data are solid.

Reviewer #3: I am satisfied with the work performed by the authors to address my concerns. My first comment was poorly worded (my apologies); I meant to ask if -65+/- 6 mv and -59+/-7 mv were actually significantly different, given these are mean+/-SEM values. I admit that I still would have liked to see a recording of APs from mutant fish but I accept their suggestion of including data in S2, S3 ansd S4.

---

## [Editor Report · Decision Letter 3]

22 Oct 2025

Dear Dr Allard,

Thank you for the submission of your revised Short Reports "Zebrafish fast muscle contractions avoid the mammalian requirement for voltage-gated Na+ channels" for publication in PLOS Biology. On behalf of my colleagues and the Academic Editor, Simon Hughes, I am pleased to say that we can in principle accept your manuscript for publication, provided you address any remaining formatting and reporting issues. These will be detailed in an email you should receive within 2-3 business days from our colleagues in the journal operations team; no action is required from you until then. Please note that we will not be able to formally accept your manuscript and schedule it for publication until you have completed any requested changes.

PRESS

Sincerely,

Taylor 

Taylor Hart, PhD,

Associate Editor

PLOS Biology

thart@plos.org